# Development of an RT-qPCR Assay for the Detection of an Emerging Duck Egg-Reducing Syndrome

**DOI:** 10.3390/vetsci12030241

**Published:** 2025-03-03

**Authors:** Zhifei Zhang, Xin Su, Dun Shuo, Dawei Yan, Xue Pan, Bangfeng Xu, Minghao Yan, Shuxuan Ren, Qinfang Liu, Chunxiu Yuan, Qiaoyang Teng, Zejun Li

**Affiliations:** Department of Avian Infectious Diseases, Shanghai Veterinary Research Institute, Chinese Academy of Agricultural Sciences, Shanghai 200241, China; nzhangzhifei@163.com (Z.Z.); suxin900220@163.com (X.S.); dun.shuo@hotmail.com (D.S.); yandawei@shvri.ac.cn (D.Y.); panxue@shvri.ac.cn (X.P.); xubangfeng@shvri.ac.cn (B.X.); m17519478956@163.com (M.Y.); r1050300191@163.com (S.R.); liuqinfang@shvri.ac.cn (Q.L.); yuanchx@shvri.ac.cn (C.Y.); tengqy@shvri.ac.cn (Q.T.)

**Keywords:** DERSV, RT-qPCR, TaqMan, specificity, sensitivity

## Abstract

Duck egg-reducing syndrome virus (DERSV), a novel Avihepatovirus, causes a gradual decline in duck egg production. Early detection of infected ducks is essential for preventing the spread of the virus and minimizing its economic impact. This study developed a quantitative reverse transcription PCR (RT-qPCR) assay for DERSV detection, which was validated for high sensitivity, specificity, and reliability. This method offers significant potential for use in epidemiological and pathogenesis studies.

## 1. Introduction

Since 2016, frequent outbreaks of egg-reducing syndromes, linked to an unidentified virus, have caused significant economic losses in duck farms across China. In 2022, Su et al. identified the virus as a new species within the Avihepatovirus genus of the picornavirus family, naming it the duck egg-reducing syndrome virus (DERSV) [1]. Picornaviruses are small, spherical, non-enveloped positive-stranded RNA viruses [2]. The genomes range from 7000 to 10,000 nucleotides in length. The RNA is linked to a small peptide at the termination of the 5′ untranslated region (UTR) and a poly (A) tail at the end of the 3′ UTR [3]. The genome contains a single long open reading frame (ORF) encoding a polyprotein, and viral proteases processed it into structural and nonstructural proteins [4]. As of early 2022, the Picornaviridae family consists of 63 officially recognized genera, with the family rapidly expanding [5]. Many picornaviruses are significant pathogens in both humans and animals, including hand-foot-and-mouth disease, poliomyelitis, and various respiratory infections, causing diseases that affect the central nervous system, respiratory and gastrointestinal tracts, liver, pancreas, heart, eyes, and skin [6,7]. Until now, the picornaviruses that have an important impact on birds include duck hepatitis A virus (DHAV) [8] and avian encephalomyelitis virus (AEV) [9].

As a novel picornavirus, DERSV infection is primarily characterized by a gradual decline in laying rate, dropping from a peak of 90% to 50%. The laying rate eventually recovers after some time, and infected ducks do not die. The virus is transmitted through direct contact among ducks. Infected individuals exhibit liver and kidney hemorrhage, follicular hemorrhage, and follicular rupture. Additionally, viral antigens are detectable in the liver, ovaries, and kidneys, where pathological changes are observed [1].

Real-time PCR is a well-established approach for quantification and detection of virus. To date, several real-time PCRs have been established, and the most commonly used real-time PCR chemistries are TaqMan probes and SYBR green [10]. TaqMan real-time PCR could template quantification and high-throughput screening with high sensitivity and precision [11], so it has been used in many clinical laboratories for virus pathogenesis studies and epidemiological investigations [12,13,14]. We have established conventional RT-PCR and nested RT-PCR of DERSV, but they are cumbersome and time-consuming, which also lacks quantitative assays. The aim of this study was to develop a TaqMan probe-based RT-qPCR assay targeting the 3D gene of DERSV to quantify viral load. The assay’s applicability was evaluated for detecting DERSV in clinical samples.

## 2. Materials and Methods

### 2.1. Viral Strains and Clinical Samples

DERSV (strain AH204), duck Tembusu virus (DTMUV), H9N2 influenza virus (H9N2 AIV), duck reovirus (DRV), and duck hepatitis viruses type 1 and 3 (DHAV-1 and DHAV-3) were isolated and stored in our laboratory. A total of 153 archived tissue homogenates (from brain, pharynx, lung, heart, proventriculus, liver, spleen, pancreas, oviduct, follicles, and colon) were collected from sick ducks in Shandong province between 2016 and 2023.

### 2.2. Primers and Probe Design

The gene sequences of DERSV strains were referenced from GenBank and aligned using DNASTAR software (DNASTAR, Madison, WI, USA). Since the 3D gene is highly conserved among DERSV strains, it was chosen as the target for RT-qPCR analysis. Primers and a probe were designed using the IDT online server (Table 1) and were synthesized by GENEWIZ (Suzhou, China).

### 2.3. RNA Extraction and cDNA Synthesis

Viral RNA was extracted using the TIANamp Virus RNA Kit (Tiangen, Beijing, China) following the kit’s protocol, and viral RNA was reverse transcribed into cDNA using Vazyme M-MLV (H^−^) Reverse Transcriptase (Vazyme, Suzhou, China) according to the manufacturer’s guidelines. The RNA and cDNA samples were stored at −80 °C until further use.

### 2.4. Construction of Standard Plasmid

A part of the DERSV 3D gene was amplified using the primers mentioned before. The PCR product was embedded into the pMD19-T vector with the Ampicillin resistance gene and transformed into DH5α *E. coli* cells. Clones containing the 3D gene were sequenced using M13 F/R primers. The recombinant pMD19T-3D plasmid was purified using the TIANprep Mini Plasmid Kit (Tiangen, Beijing, China) according to the kit’s protocol, and the concentrations were quantified using an ND-2000 spectrophotometer (Thermo Fisher Scientific, Waltham, MA, USA). The recombinant plasmid’s copy numbers were calculated according to the methods described previously [15]. The pMD19T-3D plasmid was 10-fold serially diluted in TE buffer (10 mmol/L Tris-HCl, 1 mmol/L EDTA) to concentrations ranging from 10^9^ to 10^1^ copies/μL and stored at −80 °C until further use.

### 2.5. RT-qPCR Assay

The concentrations of the primers and probe were optimized using a matrix approach. RT-qPCR was performed in a 20 μL reaction mixture containing 10 μL of 2 × PCR buffer and 1 μL template. Primers (10 μmol/L), probe (10 μmol/L), and sterile water were added to the reaction with different volumes to optimize the assay. Amplification and detection were carried out using an ABI Q5 instrument (Applied Biosystems, Foster City, CA, USA) following these conditions: 95 °C for 2 min, followed by 40 cycles of amplification at 95 °C for 20 s and 54 °C for 1 min.

### 2.6. Specificity Analysis

The specificity analysis was performed using viral cDNA. The RT-qPCR assay was performed to amplify pathogens including DTMUV, H9N2 AIV, DRV, DHAV-1, and DHAV-3.

### 2.7. Sensitivity and Repeatability of the RT-qPCR Assays

The plasmid pMD19T-D3, with concentrations ranging from 10^9^ to 10^1^ copies/μL, was used for the sensitivity assay. To assess the repeatability of the RT-qPCR, 10^7^, 10^6^, and 10^5^ copies/μL of pMD19T-D3 were tested in quintuplicate on five separate occasions to calculate the coefficient of variation (CV). Both intra- and inter-assay CVs for Ct values were determined.

### 2.8. Detection of the Clinical Sample

A total of 153 clinical samples were collected from ducks suffering from duck egg-reducing syndrome. Viral RNA was extracted and detected using the developed RT-qPCR assay. The RT-PCR method we have established [16] was used to test these samples, and the RT-PCR products were sequenced (GENEWIZ, Suzhou, China). The clinical samples were also analyzed for the presence of DTMUV [17], AIV [18], DHAV [19], Duck enteritis virus (DEV) [20], and DRV [21].

## 3. Results

### 3.1. Optimization for the RT-qPCR

The optimal concentrations of primers and probe, which resulted in the highest fluorescence and the lowest threshold cycle, were determined as follows: 0.4 μmol/L for both primers and 0.2 μmol/L for the probe. The optimized reaction system was a 20 μL reaction mixture containing 10 μL of 2 × PCR buffer, 0.8 μL of each primer (10 μmol/L), 0.4 μL of probe (10 μmol/L), and 1 μL of template.

### 3.2. Standard Curve

The pMD19T-3D plasmid was 10-fold serially diluted in TE buffer, and a standard curve was generated from 10^2^ to 10^9^ copies (Figure 1). The assay demonstrated linearity over a 10^8^ dilution range of template DNA, with a reaction efficiency of 92.59% and an R^2^ value of 0.999.

### 3.3. Specificity of the RT-qPCR Assay

Using the viral cDNA of DTMUV, H9N2 AIV, DRV, DHAV-1, and DHAV-3 as a template, only DERSV was detected by RT-qPCR while other viruses were not detected (Figure 2), demonstrating the assay was high specifically.

### 3.4. Sensitivity and Repeatability of the RT-qPCR Assay

A dilution series of the pMD19T-3D plasmid, ranging from 10^9^ to 10^1^ copies/μL, was used as the template. The RT-qPCR assay demonstrated a detection limit of 10^2^ copies (Figure 3). Three independent replicates were conducted, all yielding consistent results. The intra-assay CVs of 10^7^, 10^6^, and 10^5^ copies of pMD19T-3D were 0.27%, 0.38%, and 0.44%, and the inter-assay CVs were 0.32%, 1.85%, and 0.88%, respectively (Table 2).

### 3.5. Clinical Sample Test Results

Of the 153 samples evaluated, 72 were positive for DERSV RNA, resulting in a positivity rate of 47.06% (72/153). Ten samples were positive for AIV and negative for DERSV. None of the samples exhibited detectable levels of DTMUV, DHAV, DEV, or DRV. Among the 72 DERSV positive samples, the liver, spleen, lungs, and proventriculus exhibited positive results in over 40% of the specimens. The Ct values for these samples ranged from 22.27 to 29.66. The samples positive for DERSV by RT-PCR were confirmed by sequencing of the amplicons.

## 4. Discussion

DERSV is a recently discovered virus belonging to the Avihepatovirus genus within the Picornaviridae family, first detected in ducks in 2016. Its pathogenic mechanisms and molecular epidemiology remain unexplored. Although we have developed RT-PCR and nested RT-PCR methods for detecting DERSV, these methods are time-consuming and do not allow for detailed quantitative analysis. In contrast, real-time PCR has become widely accepted due to its speed, simplicity, and reproducibility, while also minimizing the risk of carry-over contamination compared to conventional PCR [22].

Currently, a variety of qPCR methods have been developed for the detection of viruses belonging to the Picornaviridae family. Huang, Qiuxue et al. established a real-time quantitative PCR assay for detecting duck hepatitis A virus, and the limit of detection was 3.36 × 10^3^ copies [23]. Li, Yeqiu et al. developed a duplex RT-PCR assay for the detection of duck hepatitis A virus (DHAV-1) and duck astrovirus type 3 (DAstV-3), which is a valuable tool for the detection of coinfection [24]. Liu, Qingtian et al. developed a SYBR Green based RT-qPCR assay for avian encephalomyelitis virus detection, and the sensitivity was 100 times more compared with the conventional RT-PCR method [25]. All these studies show that real-time PCR is a valuable technique for clinical diagnosis and detection.

In this study, we developed a TaqMan-based RT-qPCR assay for detecting DERSV infection, which demonstrated high sensitivity, specificity, and good repeatability. The results showed that the standard curve exhibited a linear correlation (R^2^) of 0.999 and an efficiency of 92.56%. The assay exhibited no cross-reactivity with 6 duck-derived pathogens. In future studies, additional pathogens, such as Circovirus and Paramyxovirus, should be examined to evaluate the specificity and establish a foundation for the development of multiplex RT-qPCR. The detection limit for DERSV was 10^2^ copies/μL, and the intra-assay and inter-assay coefficients of variation were 0.44% and 1.85%, respectively. DERSV was detected primarily in the lung, proventriculus, liver, and kidney, suggesting that these organs should be prioritized in clinical investigation. These findings demonstrate that this qPCR assay is a reliable and reproducible platform.

To date, we have isolated 9 strains of DERSV from different provinces, which exhibit high amino acid similarity and certain nucleotide variations [1]. These strains are capable of infecting different duck strains and show a trend of gradual spread. Additional sample collection is essential to monitor the spread and evolution of DERSV, providing critical data to support the development of effective strategies for its prevention and control.

## 5. Conclusions

In conclusion, the RT-qPCR assay developed for detecting DERSV RNA is sensitive, specific, and reliable. This assay could serve as a valuable tool for diagnosis, epidemiological, and pathogenesis studies.

## Figures and Tables

**Figure 1 vetsci-12-00241-f001:**
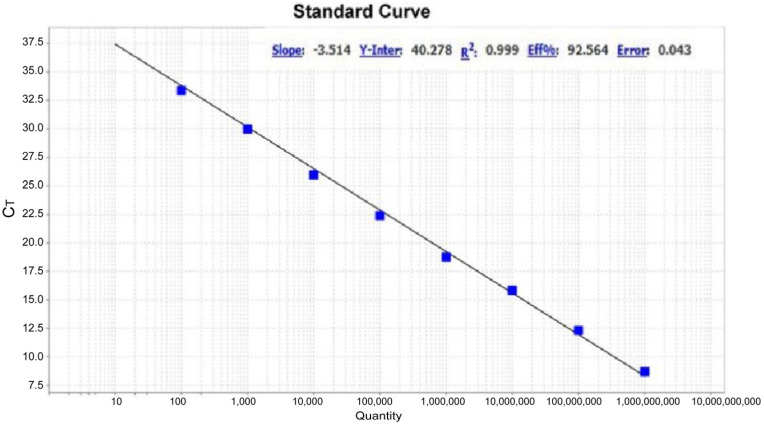
RT-qPCR assay standard curve of DERSV. A standard curve was generated using ten-fold serial dilutions of pUC19-3D, yielding a correlation coefficient of 0.999 and a slope of −3.514.

**Figure 2 vetsci-12-00241-f002:**
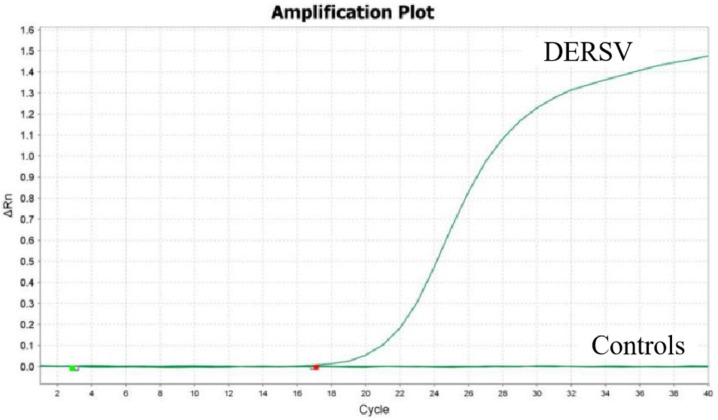
RT-qPCR assay specificity for detecting DERSV infection. Only DERSV cDNA were detected with a specific fluorescence signal. In the controls, including DTMUV, H9N2 AIV, DRV, DHAV-1, DHAV-3, and nuclease-free water, no specific fluorescence signals were detected.

**Figure 3 vetsci-12-00241-f003:**
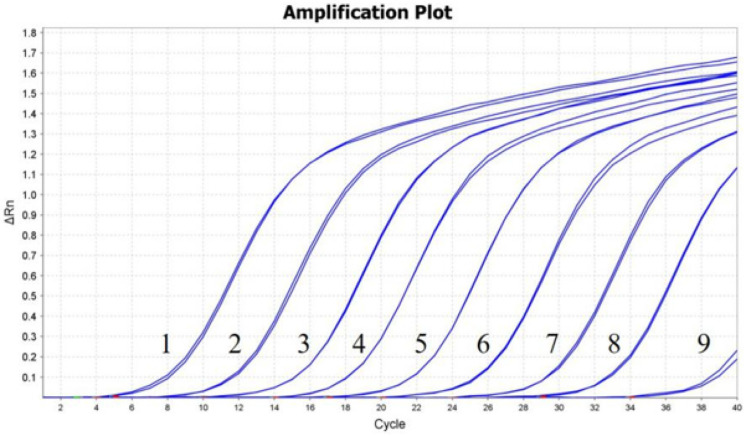
Sensitivity test for the RT-qPCR in detecting DERSV infection. Ten-fold serial dilutions of the standard pUC19-3D plasmid were amplified using the RT-qPCR assay. Amplification plots of 10^9^–10^1^ copies/uL of pUC19-3D were detected by real-time qPCR assay. Lanes 1 to 9 indicate plasmids with different concentrations ranging from 10^9^ to 10^1^ copies/uL. The lowest detectable copy number by RT-qPCR was 10^2^ copies/μL (lane 8).

**Table 1 vetsci-12-00241-t001:** Primers and probe for DERSV detection.

Gene	Primers	Sequence (5′-3′)	Position ^a^
3D	Forward	TGGGACTCAATGATGGAGAATG	7992–8013
	Reverse	TGTTTATGGAAGCAGGCTAAGA	8095–8116
	Probe	FAM-TCAAGTCATGGAGGCTGCAGTTGA-BHQ2	8069–8092

^a^ Based on GRE reference sequence (GenBank accession no. OL956952).

**Table 2 vetsci-12-00241-t002:** Intra- and inter-assay reproducibility of RT-qPCR for DERSV detection.

Concentrations of Standard Plasmid	Intra-Assay Variability		Inter-Assay Variability	
	$\bar{X}$ ± *SD*	CV(%)	$\bar{X}$ ± *SD*	CV(%)
1 × 10^7^ copies/μL	18.795 ± 0.051	0.27%	18.823 ± 0.060	0.32%
1 × 10^6^ copies/μL	22.52 ± 0.085	0.38%	22.22 ± 0.411	1.85%
1 × 10^5^ copies/μL	26.15 ± 0.114	0.44%	26.06 ± 0.229	0.88%

## Data Availability

The datasets are contained within this manuscript.

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
