# Peer review of "Development of an RT-qPCR Assay for the Detection of an Emerging Duck Egg-Reducing Syndrome"

_vetsci, 2025, doi:10.3390/vetsci12030241_

Round 1
Reviewer 1 Report
Comments and Suggestions for Authors
In this manuscript, the authors developed a RT-PCR method to detect duck egg-reducing syndrome virus (DERSV), which is a novel virus in ducks. This method is useful for the detection of DERSV in epidemiological and pathogenesis studies.
1. The viruses, DTMUV, AIV, DHAV, and DRV were used in the Specificity analysis. Some viruses such as, Circovirus and Paramyxovirus should be used. Additionally, the full name of the viruses should be listed when they were first appeared in the manuscript.
2. 153 samples were detected, and 72 of them were positive for DERSV RNA. Are these samples were specially selected with duck egg-reducing syndrome? Any other viruses were detected?
3. The application of the detection method is a key concern. Although 72 of the 153 samples were detected positive in this study, they were not confirmed by other methods to confirm the accuracy of the results. A alternative PCR should be used and sequencing of the PCR products is essential.
Comments on the Quality of English LanguageThe English language should be improved.
Author Response
Thank you very much for taking the time to review this manuscript. Please find the detailed responses below and the corresponding revisions highlighted in the re-submitted files.
Comments 1: The viruses, DTMUV, AIV, DHAV, and DRV were used in the Specificity analysis. Some viruses such as, Circovirus and Paramyxovirus should be used. Additionally, the full name of the viruses should be listed when they were first appeared in the manuscript.
Response 1: Thank you for pointing this out. I agree with your point, which also reflects a limitation of my study. I used NCBI Primer-BLAST to check the specificity of the primers and found no non-specific binding with Circovirus or Paramyxovirus gene. In future studies, I will further assess the specificity of this method against other common duck pathogens, which will lay the groundwork for developing a multiplex RT-qPCR assay.
The virus is first mentioned on page 2, lines 69-70, where I have used the full name, and I have used the abbreviation on page 3, lines 110-111.
Comments 2: 153 samples were detected, and 72 of them were positive for DERSV RNA. Are these samples were specially selected with duck egg-reducing syndrome? Any other viruses were detected?
Response 2: Thank you for pointing this out. These samples were collected from ducks suffering from duck egg-reducing syndrome and the samples' backgrounds were added on page 3, lines 120-121.
We detected common duck pathogens, such as DTMUV, AIV, DHAV, DEV, and DRV. Our results show that DERSV is the dominant virus and DERSV-positive samples were free from co-infection with other viruses.
Comments 3: The application of the detection method is a key concern. Although 72 of the 153 samples were detected positive in this study, they were not confirmed by other methods to confirm the accuracy of the results. A alternative PCR should be used and sequencing of the PCR products is essential.
Response 3: Thank you for pointing this out. Initially, we developed an RT-PCR detection method, which served as the basis for establishing the RT-qPCR assay. We used RT-qPCR to verify RT-PCR-positive samples, and the results were consistent. We also compared the sensitivity of the two methods. The detection limit of RT-PCR was 3,388 copies, while RT-qPCR had a detection limit of 100 copies.
I've already had the text reviewed and revised by a native speaker. However, I'm open to any further suggestions for improvement.
Reviewer 2 Report
Comments and Suggestions for Authors
The Editor Veterinary Sciences
Thank you for the opportunity to review the manuscript: “ Development of an RT-qPCR assay for the detection of an emerging duck egg-reducing syndrome”. The paper has been carefully reviewed but significant concerns arose:
From the moment new diseases emerge, it becomes necessary to develop new diagnostic techniques. Molecular techniques have proven to be efficient, even in samples stored for almost 10 years.
The work is presented clearly and objectively, with a compatible methodology and good results.
Line 40 – “ The RAN is linked to a small “ Correct is RNA
Author Response
Thank you very much for taking the time to review this manuscript. Please find the corresponding revisions highlighted in the re-submitted file.
Reviewer 3 Report
Comments and Suggestions for Authors
The manuscript “Development of an RT-qPCR Assay for the Detection of an Emerging Duck Egg-Reducing Syndrome” by Zhang et al is a well written, scientifically robust, and shows a great deal of advancement in veterinary virology. Developing and validating a TaqMan-based RT-qPCR assay for the detection of Duck Egg-Reducing Syndrome Virus (DERSV) would be considered a significant contribution to diagnostic virology. The present study focuses on the emerging challenges of DESRV causes in the economic loss in the duck farming industry. The specificity of the assays is clearly shown with no cross reactivity with other duck viruses with the high sensitivity and detection limit. The high positivity rate (>47%) establishes its application in the diagnostic and epidemiological investigation. I recommend acceptance of this manuscript after addition of the few insights in the discussion.
Based on the diagnostic capabilities of the method, author should elaborate a discussion how its can be use for tracking the spread and evolution of DERSV and how this model can serve for the development of the similar diagnostic tool for other emerging avian pathogen.
Author Response
Thank you very much for taking the time to review this manuscript. Please find the corresponding revisions highlighted in the re-submitted files. I elaborate a discussion on page 6, lines 186-189, and page 6, lines 191-195.